# Unsupervised learning in probabilistic neural networks with multi-state metal-oxide memristive synapses

Alexander Serb[1], Johannes Bill[2,3], Ali Khiat[1], Radu Berdan[4], Robert Legenstein[2] & Themis Prodromakis[1]

In an increasingly data-rich world the need for developing computing systems that cannot only process, but ideally also interpret big data is becoming continuously more pressing. Brain-inspired concepts have shown great promise towards addressing this need. Here we demonstrate unsupervised learning in a probabilistic neural network that utilizes metal-oxide memristive devices as multi-state synapses. Our approach can be exploited for processing unlabelled data and can adapt to time-varying clusters that underlie incoming data by supporting the capability of reversible unsupervised learning. The potential of this work is showcased through the demonstration of successful learning in the presence of corrupted input data and probabilistic neurons, thus paving the way towards robust big-data processors.

[1] Electronics and Computer Science Department, University of Southampton, Southampton SO17 1BJ, UK. [2] Institute for Theoretical Computer Science, Graz University of Technology, 8010 Graz, Austria. [3] Heidelberg University, Department of Physics and Astronomy, Kirchhoff Institute for Physics, 69120 Heidelberg, Germany. [4] Department of Electrical and Electronic Engineering, Imperial College, London SW7 2AZ, UK. Correspondence and requests for materials should be addressed to A.S. (email: A.Serb@soton.ac.uk).

Plastic synaptic connections are a key computational element of both the brain and brain-inspired neuromorphic systems. Outnumbering neurons by approximately 1,000 to 1 in the human brain[1], synapses have to perform their main function, namely interconnecting neural cells via an often modifiable coupling strength (a weight), within extremely tight volume and power budgets. The desire to build and operate large neural networks with vast amounts of synapses has rendered the task of creating similarly efficient and yet practically implementable artificial synapses a high priority.

A major route towards that goal has been the development of hardware synapse analogues, which has traditionally relied on commercially available complementary metal-oxide semiconductor technologies[2–5]. However, the visionary ideas of the early days of the field of memristor research[6,7] have led to a different approach: the exploitation of the intrinsic electrical properties of a large and diverse group of emerging nanoelectronic devices exhibiting the phenomenon of resistive switching, nowadays also referred to as memristive devices[8–10]. The scalability[11], thresholded input voltage time-integration[12], multi-level storage[13], simple two-terminal structure, potential for low power operation[14] and back-end-of-line integration[15] features demonstrated thus far in various memristive device technologies attracted study in the field of memristive synapses.

So far, the potential of memristors to act as ersatz synapses has been studied through simulation[16–21] and the demonstration of in-silico learning rule implementation, most notably—but not exclusively—that of spike timing-dependent plasticity (STDP)[22] generated by appropriate electrical memristor biasing schemes[23–27]. Other advances include the emulation of basic heterosynaptic plasticity in multi-terminal memristive devices[28], as well as the demonstration of STDP by exploiting the internal dynamics of memristors, albeit in volatile devices (that is, devices that do not retain their memory state for long periods of time, for example, 1 day)[29,30] and efforts towards the integration of memristors with neuromorphic circuits[31]. More recently the first examples of practical, small-scale artificial neural networks (ANN) operating with memristive synapses have been demonstrated, all using deterministic, supervised learning techniques. These include ref. 32, where learning was implemented using a variant of the perceptron learning rule (the Manhattan update rule), and ref. 33, where phase-change memory (PCM) rather than metal-oxide technology-based memristors are used to demonstrate learning in a Hopfield network using Hebbian learning. Finally, the first large-scale neural network using PCM technology was demonstrated by IBM[34], where a modified back-propagation rule was used in a three-layer ANN.

In this work we exploit the gradual, multi-level switching characteristics of metal-oxide-based memristors (Supplementary Note 1, including Supplementary Table 1 and Supplementary Fig. 1) for demonstrating unsupervised learning in a probabilistic neural network. Our work consolidates the current state of art in single-component synapse emulators (for example, refs 20,23,24,26,30) and advances the field of operating memristors as hardware synapse emulators in practical neural networks (for example, refs 27,32). Particularly, we demonstrate in a neural network using memristor synapses: first, pattern classification in a probabilistic neural network; second, unsupervised learning achieved through the implementation of a winner-take-all (WTA) network; third, reversible learning, an often neglected but essential aspect of truly flexible and useful learning systems and fourth, the exploitation of the intrinsic properties of our memristors to successfully allow the neural network to encode conditional probabilities without any special input signal waveform engineering.

## Results

**Weight-dependent STDP in $TiO_2$-based memristors.** STDP is one of the most widely studied plasticity rules for spiking neural networks. In its pure form it relies on the premise that the relative timing between pre- and post-synaptic spike events is the major determinant of both the direction (potentiation/depression) and the magnitude of synaptic weight changes. Recently the hardware-friendly, pulse-based biasing scheme shown in Fig. 1a–c has been proposed as a possible method for implementing STDP in memristor-based synapses[17,18,35]. The memristor's resistive state (conductance) is interpreted as the equivalent of a synaptic efficacy (weight). To implement plasticity events, the scheme exploits the inherent capability of some memristive devices to act as thresholded voltage time-integrators, that is to change their resistive state as a function of input voltage, so long as its magnitude exceeds a certain threshold (the switching threshold). When the pre-synaptic neuron spikes, a prolonged low-voltage pulse is applied across the memristor. This pulse is by itself unable to induce any resistive switching (Fig. 1a). Spiking of the post-synaptic neuron, on the other hand, leads to the application of a brief, biphasic, bipolar pulse (Fig. 1b) that causes the memristor to undergo long-term depression (LTD). Concurrent pre- and post-synaptic terminal spiking causes the memristor to

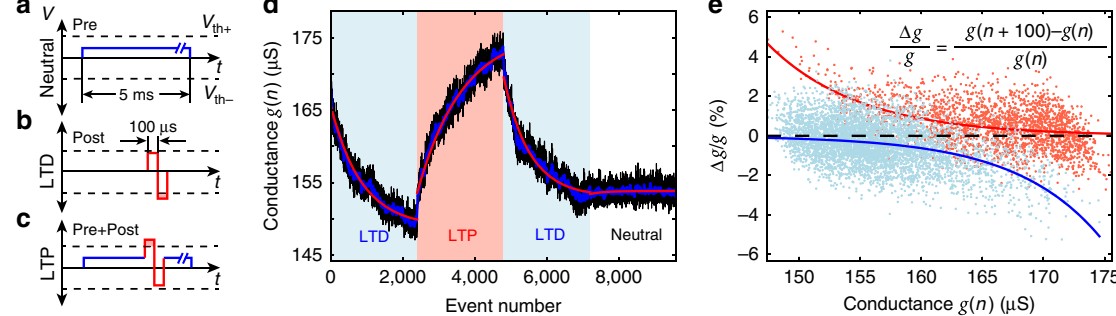

**Figure 1 | Weight-dependent STDP in memristors.** (**a–c**) Memristor electrical biasing scheme used to test STDP. $V_{th+}$, $V_{th-}$: memristor switching thresholds. Data for individual device thresholds in Supplementary Table 2. Voltage levels used to induce LTP and LTD in Supplementary Table 3. Red shading: supra-threshold portions of the input affecting the memristor resistive state. (**d**) Typical experimental results from $TiO_2$ device. Black trace: raw data; blue trace: 10-point moving average; red trace: exponential fitting. Red shading: LTP. Blue shading: LTD. No shading: neutral region, no plasticity triggered. (**e**) Experimental data and exponential fittings describing STDP magnitude (relative change in device conductance $g$) as a function of initial memristor conductance. Red line: LTP fitting. Blue line: LTD fitting. Black dashed line: zero conductance change level. Same data as in **d**.

sense the superposition of the pre- and post-synaptic spike waveforms and thereby undergo long-term potentiation (LTP; Fig. 1c).

We fabricated TiO₂-based devices (see Methods) and studied their behaviour during exposure to trains of STDP events. Each device under test (DUT) was exposed to four blocks of events, each consisting of 2,400 individual events: LTD-inducing post-only events; LTP-inducing combined pre- and post-events; LTD events again; and finally, plasticity-neutral pre-events only. Figure 1d shows typical measured results from our prototype DUTs for all mentioned electrical biasing schemes. First, we observe that the STDP rules are followed throughout the entire test, including the plasticity-neutrality of pre-only events (confirmed by experiments where pre-only events were applied at the high-conductance boundary of the DUT's operating range—Supplementary Fig. 2). Next, we observe the marked dependence of changes in resistive state on the running resistive state (DUT conductance $g$) for both LTP and LTD (Fig. 1e). Such dependence of conductance changes on the actual memristive state has commonly been observed in memristors, including both metal-oxide[25] and phase-change[36] implementations. In supervised learning rules, such as the perceptron rule, this property is undesirable as updates independent of memristive state are required[32]. Here we particularly leverage this property to enable for the first-time unsupervised learning in a practical network, in a manner similar to the work presented previously in ref. 37 that is based on simulations of PCM models.

The experimental results in Fig. 1d,e suggest that the STDP rule being implemented can be described for each plasticity event by

$$\Delta g = \text{POST} \cdot (f^+(g) \cdot \text{PRE} - f^-(g)) \qquad (1)$$

where PRE and POST are binary values indicating whether a pre- or post-spike has occurred in the given event, respectively, whilst $f^+(g)$ and $f^-(g)$ are functions that capture the influence of DUT conductance on LTP and LTD strength (also see Supplementary Note 2 and Supplementary Fig. 3). Normalizing to obtain relative changes in $g$ and rearranging we get

$$\frac{\Delta g}{g} = \text{POST} \cdot \left[ \text{PRE} \cdot f^{\text{LTP}}(g) - (1 - \text{PRE}) \cdot f^{\text{LTD}}(g) \right] \qquad (2)$$

where $f^{\text{LTD}}(g) = \frac{f^-(g)}{g}$ and $f^{\text{LTP}}(g) = \frac{f^+(g) - f^-(g)}{g}$ both fitted by exponentials in Fig. 1e.

Plotting $\Delta g/g$ versus $g$ for both LTP and LTD reveals that our solid-state synapse features inherently self-stabilizing plasticity (Fig. 1e): at higher conductance levels, further increases in conductance (LTP) become progressively smaller. Similarly, at the bottom end of the conductance scale LTD induction becomes increasingly ineffective. The gradual and monotonic dependence of weight changes on the running value of weight is an essential feature for memory models of unsupervised learning. If a stochastic data stream that triggers LTP and LTD with probabilities $p$ and $(1 - p)$, respectively, is fed into the DUT, we can expect its conductance to converge towards a unique equilibrium point. In other words, the memristive synapse should be able to encode and store in its resistive state the conditional probability $p(\text{PRE}|\text{POST} = 1)$ that a given postsynaptic spike is preceded by a presynaptic spike at the synapse within a short time interval. For instance, consider a memrisitve synapse that is exposed to STDP events that consist of a mixture of 90% LTP events and 10% LTD events. We can expect the DUT conductance to eventually stabilise close to the upper boundary of the DUT's resistive state operating range.

**Memristor synapses can encode conditional probabilities**. We experimentally tested the theoretical prediction that conditional probabilities can be encoded and stored in the resistive state of a memristor. We performed four measurement runs on the same test device. Each run consisted of 10 blocks of plasticity events ($10^4$ events per block, that is, $10^5$ events per run, blue dots in Fig. 2). Individual plasticity events were randomly chosen to be LTP events with probability $p_{\text{LTP}}$ and LTD events with probability $1 - p_{\text{LTP}}$, where the probability of an LTP event was fixed within each block. In runs 2 and 4, $p_{\text{LTP}}$ was 95%, 85%, ..., 5% for blocks 1–10, respectively, that is, the probability of LTP events was decreased after each event block. In runs 1 and 3, the same LTP probabilities were tested, but in random order (Supplementary Table 4 and Supplementary Note 3). At the end of each block the final resistive state of the memristor was measured (average of 25 read-outs after the end of each block).

The results of the experiment are shown in Fig. 2. After a burn-in phase, during which the memristor gradually reaches its normal operating range observed during the first run ($10^5$ events) we obtained consistent convergence points for the remaining three runs ($3 \cdot 10^5$ events) and a clear mapping between LTP/LTD composition and convergence conductance emerges: converged

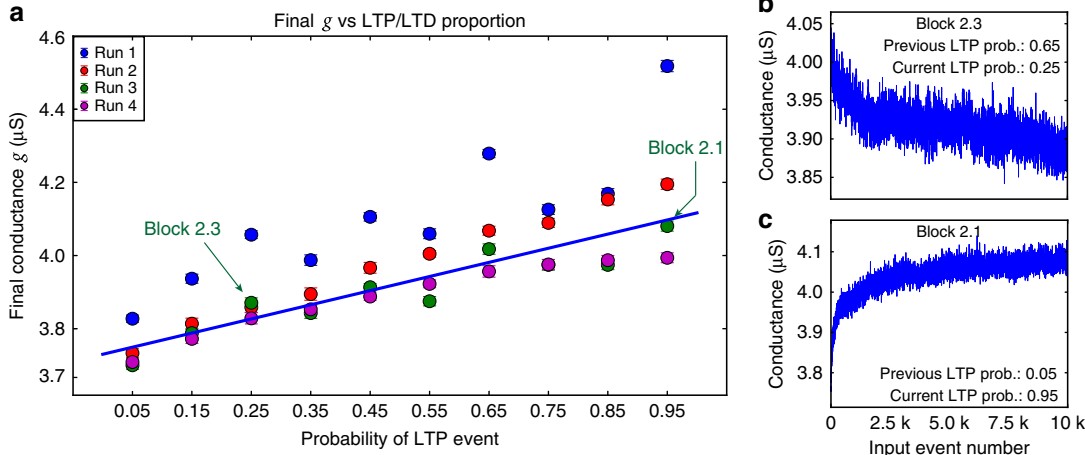

**Figure 2 | TiO₂-based memristors encode conditional probabilities.** (**a**) Final memristor conductance after application of $10^4$ input event blocks featuring different LTP/LTD compositions. Blue line corresponds to linear fit for runs 2–4. Error bars: s.d., number of samples (individual resistive state readings) per data point $n = 25$. Typical traces showing resistive state migration during two typical blocks: (**b**) one where the device is overall depressed (third block in run 2: Block 2.3) and (**c**) where it is potentiated (first block in run 2: Block 2.1).

conductance data from runs 2–4 (that is excluding burn-in) is first pooled (convergence points at each LTP/LTD composition are averaged) and then fitted to a linear function (equation and fitting parameters in Supplementary Note 4) of converged conductance versus LTP/LTD composition by least squares regression. The root mean squared error of this fitting is approximately $5.25 \cdot 10^{-2}$ μS. Moreover, we notice that the runs where the order of the LTP/LTD composition points was scrambled (1 and 3) show less well-behaved convergence points. Attempting to extrapolate memristor behaviour by exponential fitting, as presented in Supplementary Fig. 4 indicates that even $10^4$ events seem insufficient to achieve convergence given the choice of biasing parameters (Supplementary Note 5). We believe that this could be potentially addressed as more realistic memristor models appear. Thus, we can conclude that $TiO_2$ memristor-based synapses appear to be able to practically support the encoding of conditional probabilities $p(\text{PRE}|\text{POST}=1)$ in their resistive states.

**Probabilistic neural networks with memristor synapses.** The ability of individual memristors to encode conditional probabilities can be leveraged for the implementation of self-adapting spiking neural networks. In particular, WTA networks[38] have repeatedly been proposed for hardware implementations[39–42], motivated in part by the fact that WTA structures play an important role in cortical information processing[43]. Recent rigorous analyses revealed that WTA networks consisting of stochastic spiking neurons subject to weight-dependent STDP are capable of performing probabilistic inference that essentially carries out clustering of input patterns. While a number of different types of WTA networks have been considered[35,44–47], optimal parameter adaptation is in any case accomplished by weight-dependent STDP rules of the form $\Delta w \propto \text{POST} \cdot (\text{PRE} - f(w))$, that is, by rules similar to the memristor-implemented plasticity rule from equation (1).

To test whether memristor-based synapses can perform adequately as components of WTA networks, we implemented a WTA network that consisted of two stochastic spiking neurons with four inputs each. All four input synapses to one WTA neuron were implemented by $TiO_2$-based devices, while the synapses to the other neuron were implemented in software (Fig. 3a). This hybrid network allowed us to directly compare

software-simulated synaptic connections with memristive synapses in the same set-up and with exactly the same inputs. It also allowed us to directly manipulate the software synapses and study the influence on memristive plasticity.

The 2-neuron probabilistic WTA network was implemented on an in-house developed instrumentation board for memristor device characterization[48]. The two artificial neurons, WTA lateral inhibition and synapses feeding one of the neurons were all implemented in software on the board's microcontroller unit. During each experiment run 1,200 four-bit patterns were presented to the network at the inputs $\mathbf{y} = (y_0, y_1, y_2, y_3)$. Determining the values of $\mathbf{y}$ begins by randomly and equiprobably drawing a pattern to be presented from a set of prototype test patterns (in our case 0110 and 1001). Next, each bit in the selected pattern is flipped with a probability of 10% so that the network is presented with noisy instantiations of the prototype patterns. The resulting generated input vector is then multiplied by the weight vectors of both neurons and translated into membrane potential values, one for each neuron, as per equation (3):

$$U_i(\mathbf{y}, t) = \theta_i(t) + \mathbf{w}_i(t) \cdot \mathbf{y}(t) \qquad (3)$$

where $U_i(\mathbf{y},t)$ denotes the membrane potential for neuron $i$ during event $t$, $\theta_i(t)$ an adaptive excitability term that homoeostatically regulates neuron activity and $\mathbf{w}_i$ the weight vector from inputs $\mathbf{y}$ to neuron $i$. The symbol · represents the dot product operator. Importantly, whilst $U_i$ represents the membrane potential of neuron $i$ for the purposes of driving its firing behaviour, it does not directly translate to a physical voltage value to be applied to all synapse terminals (pre or post) it is connected to. Neuron firing events are instead translated into appropriate pre- or post-type voltage waveforms that are used to bias the affected memristor synapses. The homoeostatic term $\theta_i(t)$ has been used before for memristor learning[17] and has been theoretically justified in ref. 44 for unsupervised learning in probabilistic WTA networks. By reducing the propensity to fire for neurons that show high average response, homoeostasis ensures that both neurons participate in the WTA competition over the long run (details in Methods section).

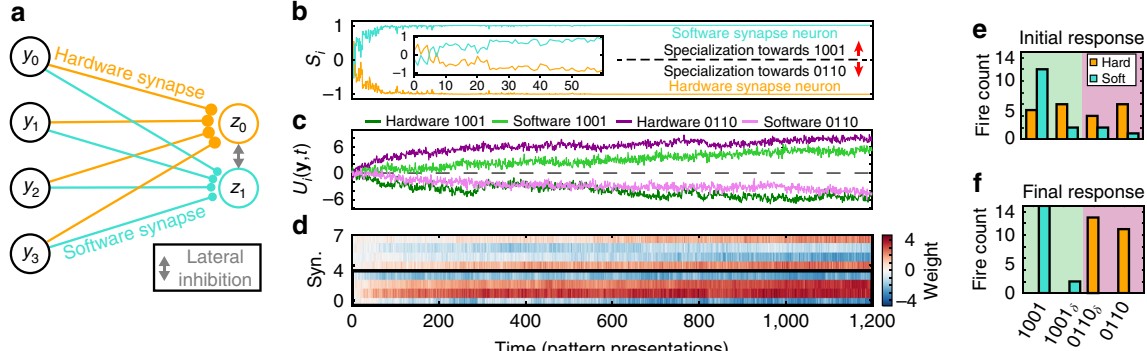

**Figure 3 | Learning in a WTA network with a mixture of software and memristor synapses.** (**a**) Diagram of the 2-neuron, WTA network used in this work. (**b**) Evolution of neuron specializations $S_i$ to patterns 0110 and 1001 as weights change over successive events, illustrating the interplay between the two neurons. Inset: close-up of first 60 trials. (**c**) Computed membrane potentials of each neuron to both prototype patterns according to their weights at every trial illustrating the intrinsic pattern preferences of each neuron, that is independent of their interaction in the WTA network. (**d**) Evolution of hardware (synapses 0–3, enclosed in thick, black frame) and software (synapses 4–7) weights. (**e**,**f**) Responses of the WTA network to the initial (**e**) and final (**f**) 41 input samples. The fire count of both the hardware synapse neuron (orange) and the software synapse neuron (turquoise) is shown for patterns 0110 and 1001, and patterns that differ from these prototypes in one position ($0110_\delta$ and $1001_\delta$). The different pattern groups are perfectly segregated by the end of the run.

The probability $p_i(\mathbf{y},t)$ with which neuron $i$ wins the WTA competition and therefore spikes at event $t$ is given by

$$p_i(\mathbf{y}, t) = \frac{e^{U_i(\mathbf{y},t)}}{\sum_j e^{U_j(\mathbf{y},t)}} \qquad (4)$$

Using computed $p_i$ values for each pattern at each time step we can define a specialization metric $S$ that directly quantifies how attuned each neuron is to the two prototype input patterns:

$$S_i(t) = p_i(1001, t) - p_i(0110, t) \qquad (5)$$

where $S_i(t)$ is the specialization of neuron $i$ at time $t$ and takes values between 1 (perfectly specialized on 1001) and $-1$ (perfect specialization on 0110).

By definition, at every event exactly one of the neurons wins and fires, thus triggering plasticity at its synapses. In the case of software synapses, weights are updated through a simple STDP rule that aims to approximately mirror memristor plasticity. The variability in resulting STDP-driven weight changes $\Delta w$ and measurement noise observed in memristor synapses have both been included in the software synapse plasticity mechanism (see Methods). In the case of the hardware synapses the STDP conditions that determine whether LTP or LTD is required are the same as for their software counterparts, but the LTP and LTD events are translated into pulse voltage stimulation and therefore the magnitude of weight change is inherently set by each memristor. For the purposes of this experiment and since the non-invasiveness of the pre-only event has already been confirmed (Fig. 1), the pulsing scheme for LTP and LTD is reduced to only the above-threshold portions of the original waveforms, that is, both LTP and LTD are represented by simple square-waves of appropriate amplitude. To map device resistive states onto weights all memristive synapses were first subjected to the protocol described in Fig. 1. Estimated maximum and minimum operational conductance values (extracted from the constant term of exponential fittings to traces in Fig. 1d—also see Supplementary Fig. 5) were mapped linearly to a weight range of $[-2.2, +2.2]$. The conductance-weight mappings are summarized in Supplementary Table 3.

Results from a WTA network experiment (run no. 1) are shown in Fig. 3. Both hardware and software synaptic weights $w_{ij}$ were initialized close to 0 (see Methods section) and subsequently the network was allowed to react to the incoming patterns freely. According to theoretical WTA models, unsupervised synaptic adaptations through STDP should lead to a clustering of inputs such that each neuron is preferentially activated by one of the prototype patterns and noisy variations of it. Figure 3 demonstrates this behaviour in our set-up with memristive synapses. The specialization evolution in Fig. 3b shows how after a brief initial phase of uncertainty where the neurons are approximately equally attuned to both patterns and none can claim dominance over either pattern (approximately first 20–30 samples), the hardware synapse neuron develops a clear preference for pattern 0110 (specialization $S$ approaches $-1$). Similarly, we can use the weights of software and hardware synapses at each trial to plot computed membrane potentials for each neuron in response to each pattern. This is shown in Fig. 3c where we observe how at the beginning of the run neither neuron has any intrinsic preference for any pattern (that is independent of the neuron–neuron interaction through the WTA); this only starts developing afterwards. The robustness of these experiments was confirmed by repeating the experiment three times in total. Results from all three runs are summarized in Supplementary Fig. 6 and Supplementary Note 6.

Examining the evolution of weight values throughout the run (Fig. 3d) we observe that the hardware synapse weights experience noisy and slow drift from their initial values. To quantify this the evolution of each weight over trials was fitted to an exponential function and the s.d. of the residual was then computed. This yielded estimates of both the noise levels and the overall weight change for each synapse over the trial (for full results see Supplementary Note 7 and Supplementary Fig. 7). The software synapses concurrently experience similarly imperfect drift towards their final state. For comparison, see Supplementary Figs 8 and 9 in the case where software synapses are noise-free. These results are confirmed by Fig. 3e,f where we see a substantially clearer classification of pattern 0110 and related patterns different from 0110 in only one position ($0110_\delta$) on the one hand (purple shading) and 1001 with $1001_\delta$ (patterns different from 1001 in only one position) on the other hand (green shading) towards the end of the experiment versus the beginning. Specifically, at the beginning of the run patterns 1001 and $1001_\delta$ cause the neuron that ultimately assigns itself to them (software synapse) to fire only approximately 56% of the time whilst similarly the hardware synapse neuron responds to its corresponding patterns (0110 and $0110_\delta$) approximately 77% of the time. In contrast, at the end of the run classification accuracy increases to 100% for both neurons. Thus, the WTA network successfully segregates the prototype patterns despite the presence of noise. This result was achieved in a fully unsupervised manner. An example case of how the same test evolves when software synapse imperfections are suppressed is shown in Supplementary Note 8 and Supplementary Figs 8 and 9.

Finally, to demonstrate that the WTA network is capable of not only learning a pattern but also if demanded forgetting and relearning it, a further set of experiments was conducted. This consisted of two further, consecutive WTA learning runs (runs no. 2 and 3) immediately following the main run from Fig. 3 (by the end of which we recall the memristor synapses had specialized their neuron to pattern 0110). At the beginning of each of these additional runs the software synapses were initialized such that the network specialization acquired during the immediately preceding learning run was reversed (hardware synapses were left unchanged). Under these circumstances the memristor-based synapses are expected to respond by flipping their intrinsic preference to the opposite pattern. Results are shown in Fig. 4.

In the case of the first additional run, the software synapses were initialized in such way as to instantly reverse the preferred pattern-to-neuron mapping outcome of the previous learning session and start the learning run with the software synapse, rather than the memristor synapse neuron more responsive to pattern 0110. Such initialization should induce the memristor synapses to attempt specializing on pattern 1001 instead. The top half of Fig. 4 shows that this is indeed the case: at the end of the run the hardware synapse neuron has lost its intrinsic preference to pattern 0110 and began switching to 1001 as evidenced by the membrane potential plot (Fig. 4b), which allowed the software neuron to consolidate its dominance of 0110 (Fig. 4a). Simultaneously, the software synapse weights remain relatively static around their extreme values, as initialized. The second additional run similarly initializes the software synapses appropriately to guide the memristor synapses to re-specialize on pattern 0110. This successfully occurs as evidenced by Fig. 4f–j and confirmed by additional runs shown in Supplementary Fig. 10 and Supplementary Note 9. In both cases, the fire count histograms (Fig. 4d,e,i,j) show how the initial classification preferences of each neuron become entrenched during each run as a result of the combined changes in both software and hardware synapse weights with hardware synapses mainly driving the process (Fig. 4b,g and Supplementary Table 5).

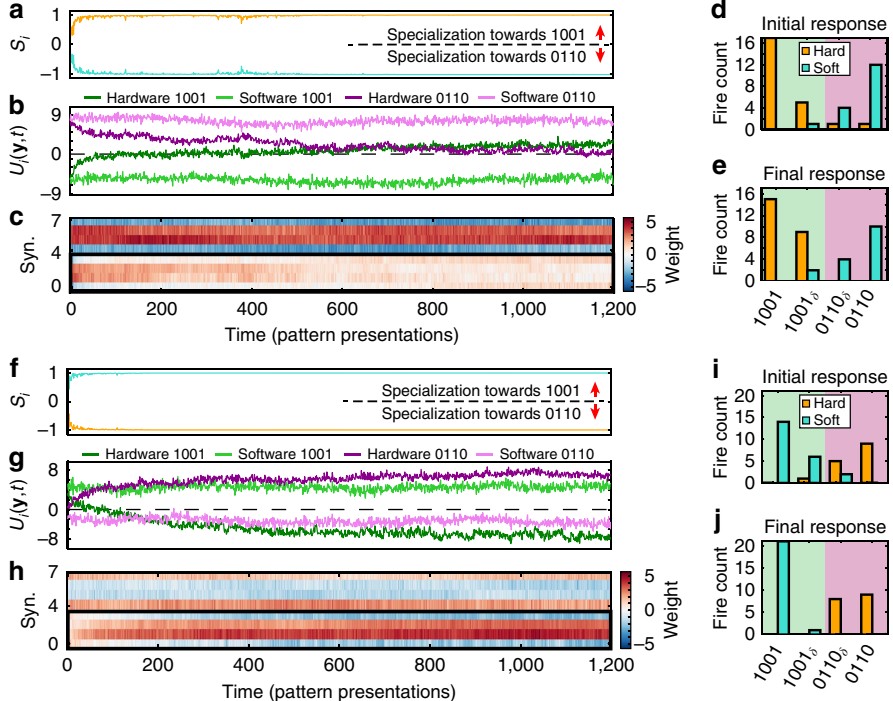

**Figure 4 | Reversible learning is supported in WTA networks using TiO$_2$ memristor-based synapses.** (**a–e**) First run attempting to unteach the pattern recognition abilities gained in Fig. 3. (**a**) Evolution of neuron specializations $S_i$ to patterns 0110 and 1001 as weights change over successive events, illustrating the interplay between the two neurons. (**b**) Computed membrane potentials of each neuron to both prototype patterns according to their weights at every trial illustrating the intrinsic pattern preferences of each neuron, that is independent of their interaction in the WTA network. (**c**) Evolution of hardware (synapses 0–3, enclosed in thick, black frame) and software (synapses 4–7) weights. (**d,e**) Responses of the WTA network to the initial (**d**) and final (**e**) 41 input samples. The fire count of both the hardware synapse neuron (orange) and the software synapse neuron (turquoise) is shown for patterns 0110, 1001 and patterns that differ from these prototypes in one position (0110$_\delta$ and 1001$_\delta$). (**f–j**) Corresponding data as in **a–e** for second run attempting to reteach the memristor synapses to prefer pattern 0110. The abrupt changes between final and initial responses over consecutive experiments mainly arise from the different initializations of the software synapses in each case.

## Discussion

In this work we demonstrated that metal-oxide-based synapses with inherent, gradual, self-limiting switching properties are capable of learning and re-learning of input patterns in an unsupervised manner within a probabilistic WTA network. Key to the learning process is the memristors' capability of encoding conditional probabilities of the expected input signal within their resistive states. As a notable consequence of the probabilistic learning scheme, ubiquitous (and unavoidable) noisy changes in the resistive states are continuously counterbalanced by the ongoing alignment of present weights with future presented inputs.

This study was performed on TiO$_2$-based devices, which has historically been one of the significant metal-oxide systems used in memristive devices[49]. In previous work, we have identified that these devices support multi-level switching[50], the emulation of short- and long-term plasticity[20,30], and bidirectionally gradual switching[51], which we can reliably detect using our tailor-made instruments[48] even at low OFF/ON resistive state ratios. A brief discussion on the electrochemistry behind our devices is included in Supplementary Note 10. Endurance and retention data on our devices are shown in Supplementary Figs 11 and 12. Here we build on our previous results for demonstrating a memristor-based, system-level application. The presented concept may extend to other memristor technologies based on different metal-oxides such as HfO$_2$ and Ta$_2$O$_5$ that have shown great promise towards memory applications.

For the purposes of this work, our prototypes were operated under low voltage conditions, that is close to their threshold voltages (Supplementary Tables 2, 3). Importantly, the devices' threshold voltages are not rigidly fixed, but rather depend on stimulus waveform shape, as well as the initial memory state of the devices. For example, the threshold voltage dependence on square-wave pulse duration is shown in the Supplementary Fig. 13. As a result, the voltage amplitude of the pre-waveform, as shown in Fig. 1a–c, is important as it determines the voltage contrast between: First, the super-threshold peak in the pre + post waveform and the sub-threshold peak in the post-only waveform and second the pre-only waveform and the post-only peak. Larger contrasts mean that spurious drift effects induced by threshold voltage variability can be mitigated more effectively. This reduces the risks arising from unwanted plasticity caused by repeated pre-pulsing without any post-response, as well as unwanted, concurrent DUT resistive state disturbance by both the peak and the trough of the post-waveform.

Considering future implementations of practical memristor-based systems we note the following: First, the downscaling of the memristor component itself as a memory storage element is already comparing favourably to mainstream technologies (for example, static random access memory—SRAM), as memristors in the 10 nm × 10 nm = 100 nm$^2$ range have already been demonstrated[11]. SRAM scaling is projected to become difficult at below 50,000 nm$^2$ even under favourable process variability conditions[52] (but note 1T-SRAM technology[53]). Even though the performance of memristor devices is also known to be impacted by downscaling, through, for example, increased access wire resistance, the advantage over SRAM is expected to dominate. Furthermore, we note that memristors can pack

more than 1 bit/storage element in a non-volatile manner whilst SRAM is purely digital and volatile. Second, at the array level, the packing density of memristors can be in-principle increased by the development of high density three-dimensional crossbar arrays[54], where back-end integrable selector elements[55] could mitigate the well-known sneak path problem[56]. Third, at the peripheral circuit level the trade-off between memristor functionality and circuit complexity needs to be studied more in-depth. Standard square pulse generators (for write) and sense amplifiers (for read), also used in conventional memory systems, might suffice if memristors are to be treated as binary data storage elements. More complicated circuits capable of generating multiple voltage levels (write) and reading absolute resistance values will be, however, needed for multi-state operation; a compromise between higher bit resolution operation and required silicon real estate for peripheral circuits. Finally, the challenges of interfacing with analogue hardware-based artificial neurons have to be considered. Optimized operation will be achieved if the artificial neurons output spikes of the forms exhibited in Fig. 1a–c and all the voltages involved are within the headroom required by the artificial neuron circuitry. If the former condition is not met, then each neuron will need to be equipped with a suitable output waveform-shaping circuit at the moderate cost of 1 per neuron. This can be expected to be a relatively minor inconvenience if the waveforms involved are simply variable duration square waves; easily obtainable via digital clock signals. If the latter condition is not met, then additional supply rails will have to be introduced on-chip and the output waveform-shaping circuits will require level shifters of voltage difference-related levels of complexity; yet the cost will remain at the 1 per neuron level. Notably, in this work biasing conditions were individually tailored for each memristive synapse, a result of device-to-device variability that is expected to become increasingly challenging with downscaling. Improvements in control over fabrication and electroforming conditions are needed to counterbalance that effect and deliver memristors that operate under sufficiently uniform biasing conditions to use a single, non-programmable waveform-shaping circuit for all devices in practical systems.

The WTA architecture used in this study can be seen as a simplified version of cortical layers 2/3 where parvalbumin-positive interneurons provide feedback inhibition to pyramidal cells (see, for example, refs 44,46,47,57,58 for similar models). Recent experimental data on the connectivity dynamics in cortical circuits suggest that synaptic modifications in the cortex are stochastic (for example, refs 59–61). This is of particular relevance to our study as our results demonstrate WTA architectures are particularly robust against the noisy synaptic plasticity exhibited by our memristive prototypes, also noted through simulations in ref. 17. In addition, the theoretical framework introduced in refs 62,63 indicates that stochastic plasticity may even have advantageous computational properties, in that it performs Bayesian inference on optimal circuit parameters, suggesting that the inherent stochastic properties of memristors could even be beneficial to learning.

In our experiments, the prototype patterns 1001 and 0110 were presented as noisy versions where each component was independently inverted with a probability of 10%. Hence, the presented patterns for prototype 1001 included patterns 0001, 1000, 1101 and 1011. These patterns were denoted by $1001_\delta$ (analogous noisy versions of 0110 were denoted by $0110_\delta$). In particular the noisy versions 1101 and 1011 show significant overlap with the other prototype 0110 since they include one of their two non-zero bits. Our results (see, for example, Fig. 3e,f) show that the system is very robust to such pattern overlap since those neurons that specialized on the prototype also responded to the corresponding $\delta$ patterns after learning. For the current

set-up, we did not use pattern overlap in the prototype patterns because of their very low dimensionality. The theory for WTA networks and experience from computer simulations (see, for example, ref. 47) show that such overlap poses no difficulties for the circuit for high-dimensional inputs. Hence, we do not expect any additional hardware cost to account for pattern overlap due to the inherent robustness of WTA circuits to such pattern sets.

In the WTA experiments, the Hebbian-type synaptic plasticity rule was complemented with a homoeostatic plasticity rule, which regulates the intrinsic excitability of the neurons. Notably, homoeostatic intrinsic plasticity only adds a bias to the neuronal membrane potential and, thus, does not affect a neuron's relative firing preference to different input patterns. It also influences the emerging synaptic weight configuration only indirectly by ensuring that all WTA neurons maintain a long-term average firing rate and thereby modulates the succession of LTP/LTD plasticity signals, which the memristor synapses observe. While homoeostatic intrinsic plasticity has been proven mathematically to harden robustness of unsupervised learning in stochastic WTA circuits, its implementation in neuromorphic designs is possible, for example, via a local accumulator circuit per neuron. Notably, homoeostatic contributions to the overall membrane potential during learning (Fig. 3) were significantly smaller than synaptic contributions as depicted in Supplementary Fig. 14.

In conclusion, in this work we have demonstrated for the first time that individual, solid-state memristors can emulate complex, weight-dependent plasticity, including unsupervised classification, forgetting and relearning, within an experimental WTA network setting. This paves the way towards real-time on-node processing of big, unstructured data; an enabling technology for addressing the challenges arising from the volume of data generated by the internet-of-things revolution.

## Methods

**Device fabrication and preparation.** For all experiments, $TiO_2$-based micro-metre-scale devices are used using a metal–insulator–metal structure. The process flow started by thermally oxidising a 6 inch Silicon wafer to create a layer that serves as an insulator medium. Then, three major steps were realized to obtain the bottom electrode, active layer and top electrode consecutively. Each step consisted of optical lithography, material deposition and liftoff process. The 10 nm platinum layers were deposited for top electrode and bottom electrode by electron beam evaporation, whilst 25 nm $TiO_2$ was deposited by reactive magnetron sputtering. These fabrication steps resulted in a metal–insulator–metal stack of $Pt(10 \, nm)/TiO_2(25 \, nm)/Pt(10 \, nm)$; devices used with slight variations for many other purposes in our group[27,51]. Before use, all devices were electroformed using positive polarity (top electrode at higher potential than bottom electrode) pulsed voltage ramps. A series resistor was used as a current-limiting mechanism in all cases. Typical electroforming voltages were in the range of 7–8 V.

**WTA network set-up.** In the WTA network, neurons fire with probability $p_i$ as determined by the abstract membrane potentials $U_i(\mathbf{y},t)$ according to equations (3) and (4). The network response in turn triggers plasticity of the hardware and software synapses, as well as of the excitabilities $\theta_i$. For the WTA network set-up, we hence have to define three quantities: the plasticity rule of software synapses; a function that maps the memristor conductance values $g$ to abstract weights $w$ in equation (3) (conductance to weight map function); and the plasticity rule of the excitability $\theta_i$. The plasticity rule of hardware synapses is inherently controlled by the memristors.

For software synapses $w_{ij}$ we fundamentally use a plasticity rule of the form

$$\Delta w_{ij} = \eta \cdot POST \cdot \left(PRE - f(w_{ij})\right) \qquad (6)$$

where the learning rate $\eta = 0.03$. The weight-dependent function $f(w_{ij})$ will be determined such that it approximately mirrors the plasticity of memristor synapses. The structure of $f(w_{ij})$ can be estimated from the measured memristor plasticity in

Fig. 1e. Using equations (1) and (2) we find

$$\Delta g \propto \text{POST} \cdot \left( \text{PRE} - \frac{f^-(g)}{f^+(g)} \right)$$

$$= \text{POST} \cdot \left( \text{PRE} - \frac{\frac{f^-(g)}{g}}{\frac{(f^+(g)-f^-(g))}{g} + \frac{f^-(g)}{g}} \right) = \qquad (7)$$

$$= \text{POST} \cdot \left( \text{PRE} - \frac{f^{\text{LTD}}(g)}{f^{\text{LTP}}(g) + f^{\text{LTD}}(g)} \right) \qquad (8)$$

Supplementary Fig. 3 shows the fraction on the right-hand side of equation (8) based on the fitted functions $f^{\text{LTP}}(g)$ and $f^{\text{LTD}}(g)$ from Fig. 1e. As can be seen in the figure, the measured plasticity curves of the memristor suggest a sigmoidal shape for the function $f(w_{ij})$ in equation (6). This observation can be substantiated analytically: by inserting the exponential fits $f^{\text{LTP}}(g) = \exp\left(-\frac{1}{2}\alpha_{\text{P}} \cdot (g - \beta_{\text{P}})\right)$ and $f^{\text{LTP}}(g) = \exp\left(\frac{1}{2}\alpha_{\text{D}} \cdot (g - \beta_{\text{D}})\right)$ into equation (8), a few lines of algebra yield

$$\Delta g \propto \text{POST} \cdot \left( \text{PRE} - \sigma \left( \frac{\alpha_{\text{P}} + \alpha_{\text{D}}}{2} \cdot \left[ g - \frac{\alpha_{\text{P}}\beta_{\text{P}} + \alpha_{\text{D}}\beta_{\text{D}}}{\alpha_{\text{P}} + \alpha_{\text{D}}} \right] \right) \right) \qquad (9)$$

where we define $\sigma(x) = (1 + \exp(-x))^{-1}$. Such sigmoidal shape was qualitatively observed for all memristor synapses (as seen in Supplementary Fig. 5), which served as a reference for the shape of software plasticity. On the basis of the comparison of equations (6) and equation (9), we map memristor conductances $g$ to abstract weights $w$ via a linear function

$$w = \alpha \cdot (g - \beta) \qquad (10)$$

and set $f(w_{ij}) = \sigma(w_{ij})$ in equation (6), thereby tackling the software synapse plasticity rule and the conductance to weight function.

**Adding realistic imperfections in software synapse function.** On top of this ideal, theoretical framework we have added two mechanisms of software synaptic weight corruption to better match the memristors' own noisy and variable behaviour. Under this more realistic framework we make a distinction between the true, underlying weight $w_{ij}$ and the as measured weight, including measurement noise $v_{ij}$. The first weight corruption mechanism reflects the memristors' cycle-to-cycle variation, which in our case manifests itself as variable conductance jumps given identical stimulus and initial conductance conditions. This is modelled by adding a switching variability term $w_{\text{var}}$ drawn from a Gaussian distribution with $\sigma_{\text{sw}} = 0.04$ (units of abstract weight) limited to $\pm 5\sigma$. The weight update equation thus becomes

$$\Delta w_{ij} = \eta \cdot \text{POST} \cdot \left[ (\text{PRE} - f(w_{ij})) + w_{\text{var}} \right] \qquad (11)$$

where $\sigma_{\text{sw}}$ was chosen to qualitatively force the software synapses to show slightly worse cycle-to-cycle variation than what was being observed in the hardware. This is evidenced in the Supplementary Fig. 7, where the evolution of individual synaptic weights during an ANN learning trial is plotted.

The second weight corruption mechanism introduces a degree of measurement noise in the software synapses, that is, allows the system to use a slightly distorted weight value without causing any change in the underlying value of $w_{ij}$. As such, at every time step, the weight values used to compute neuron membrane potentials and by extension contribute to deciding, which neuron fires to each presented input are calculated by the following formula:

$$v_{ij} = w_{ij} + w_{\text{mn}} \qquad (12)$$

where $w_{\text{mn}}$ is an added measurement noise term drawn from a Gaussian distribution with $\sigma_{\text{meas}} = 0.4$ (abstract weight), limited to $\pm 5\sigma$. $\sigma_{\text{meas}}$ was determined by estimating/quantifying the measurement noise in our devices and adjusting the software so as to behave slightly more stochastically than the memristors (Supplementary Table 6).

**Homoeostatic plasticity.** Furthermore, to facilitate robust learning we use a homoeostatic plasticity mechanism for the excitabilities $\theta_i$. At the beginning of each learning experiment (initial learning only, this does not apply to reversibility learning experiments where continuity of $\theta_i$ is maintained), the $\theta_i$ are initialized at 0. Then, before each time step $t$ the excitability is updated according to

$$\theta_i(t) = \begin{cases} \theta_i(t-1) - \eta_\theta/2 & \text{if neuron } i \text{ wins event } t-1 \\ \theta_i(t-1) + \eta_\theta/2 & \text{otherwise} \end{cases} \qquad (13)$$

with learning rate $\eta_\theta = 0.03$. The homoeostatic plasticity rule (13) makes sure that both neurons will participate in the competition and fire, on average, equally often: if a neuron fires on average during one half of the time steps, the value of its $\theta_i$ will remain approximately stable. Otherwise, its $\theta_i$ will slowly increase (if the neuron fires rarely) or decrease (if the neuron fires frequently). The rule (13) defines the plasticity rule of the excitability. Notably this homoeostasis rule is very similar to the one used in ref. 17 where although specific details are not given, the spiking frequency of all neurons is periodically assessed and an equivalent to the $\theta_i$ term is adjusted accordingly. In this work this procedure takes place at every trial, which may allow finer and more responsive homoeostatic control. The behaviour of this plasticity rule is described in ref. 44.

**Table 1 | Key ANN operating parameters.**

| Symbol | Value | Units | Parameter |
|---|---|---|---|
| $\eta$ | 0.03 | — | Synaptic weight learning rate |
| $\eta_\theta$ | 0.03 | — | Homoeostatic plasticity learning rate |
| $\sigma_{\text{sw}}$ | 0.04 | Abstract weight | Switching noise (software synapse) |
| $\sigma_{\text{meas}}$ | 0.4 | Abstract weight | Measurement noise (software synapse) |

**Memristor parameter extraction.** For the WTA experiment, the parameters in equation (10) must be individually determined for each memristor. To this end, the conductance operating range of each device was extracted in the set-up of Fig. 1d before the WTA experiment. The parameters $\alpha$ and $\beta$ were then implicitly defined by directly mapping two conductance points $g_{\text{LOW}}$ and $g_{\text{HIGH}}$ to abstract weight values $-2.2$ and $+2.2$, respectively. The values for $g_{\text{LOW}}$ and $g_{\text{HIGH}}$ for each device are shown in Supplementary Table 3. The numerical values for all initial and final weights during the WTA experiments are provided in Supplementary Table 5 for both software and memristive synapses.

**Network initialization procedures.** The experimental run corresponding to Fig. 3 (and similar, confirmation runs included in the Supplementary Material) required all weights to be initialized as close to 0 as possible. For the hardware synapses this was done through the memristor-handling instrument (Supplementary Fig. 15) by manually applying a suitable number of square wave pulses on each device. We did not seek to automate this process at this stage. For the software synapses, the initial underlying weights $w_{ij}$ were set to 0, but then corrupted by measurement noise before use as described above. A summary of the key network operating parameters is given in Table 1.

**Instrumentation.** All experiments were carried out using an upgraded version of the in-house instrumentation described in ref. 48. More details provided in Supplementary Note 11.

**Data availability.** All data supporting this study and its findings are available within the article, its Supplementary Information and associated files. This includes relevant software code. Any source data deemed relevant is available from the corresponding author on request.

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

## Acknowledgements

This work has been supported by the Engineering and Physical Sciences Research Council (EPSRC) grants EP/K017829/1 and EP/J00801X/1, the Austrian Science Fund FWF grant #I753-N23 and the CHIST-ERA ERA net PNEUMA project.

## Author contributions

The first two authors contributed equally to this work by setting up and performing the experiments; A.K. fabricated the devices; R.B. assisted with the experiments and instrument preparation; T.P., R.L., J.B. and A.S. conceived ideas and have written the manuscript; T.P. and R.L. are joint last authors.

## Additional information

**Competing financial interests:** The authors declare no competing financial interests.

