## [Peer Review File · Nature Communications]

Reviewer Comments

Reviewer #1 (Remarks to the Author):

A. Summary of the key results

In this work, Serb and coauthors investigate the use of memristors as artificial synapses that learn using a biology inspired learning rule (STDP). The paper associates theoretical analysis, experiments and simulations in a convincing and rare way. In particular, a small system is investigated experimentally.

B. Originality and interest

Learning with memristive synapses has already been done, but with simpler learning rules (eg Ref 24). Using the more sophisticated STDP is an idea that interests the field considerably. STDP has been done experimentally with memristors only in simplistic visions on single devices. Much deeper studies have been done, but in simulation only. This is the first work that validates experimentally, on a very simple system, the use of STDP with memristors. The comparison with simulations is insightful: experiments appear much more noisier than simulations, but the system still learns, due to the robustness of STDP. This is what was believed would happen, but had never be shown experimentally. Of course, in this work, the experimental learning is still very modest. But I believe that the work is of high interest to the field, if some revisions are made.

The paper states that Ref 24 uses binary memristors. I believe this to be incorrect, Ref 24 is also using analogue memristors. Ref 24 is an important milestone. Instead of "a form of deterministic, supervised learning", I would prefer that the authors name the learning approach that Ref 24 is using.

C. Data & methodology: validity of approach, quality of data, quality of presentation

The paper is well written.

Figure 3 is too hard to interpret (especially Fig 3(c) that is critical to the message). Maybe the authors could split Fig 3 into several Figures?

The discussion that nonlinearity in the device is useful for unsupervised learning is important. "Such dependence

of conductance changes on the actual memristive state has commonly been observed in memristors" should be backed by references, eg:

Jo, S. H., Chang, T., Ebong, I., Bhadviya, B. B., Mazumder, P., & Lu, W. (2010). Nanoscale memristor device as synapse in neuromorphic systems. Nano letters, 10(4), 1297-1301.

The same dependence on initial state has also been observed in phase change memories, eg in:

Suri, M., Bichler, O., Querlioz, D., Traoré, B., Cueto, O., Perniola, L., ... & DeSalvo, B. (2012). Physical aspects of low power synapses based on phase change memory devices. *Journal of Applied Physics*, 112(5), 054904.

The discussion that ends in "In other words, the memristive synapse should be able to encode and store in its resistive state the conditional probability $p(P_{RE} | POST = 1)$ " is interesting and insightful. Actually, a lot of discussion about this already appears in a review paper:

Querlioz, D., Bichler, O., Vincent, A. F., & Gamrat, C. (2015). Bioinspired programming of memory devices for implementing an inference engine. *Proceedings of the IEEE*, 103(8), 1398-1416. Equations (5)-(7) of this review show exactly the points made by the authors, with a similar STDP rule. The authors should rephrase the article to take benefit of the analysis in this review.

What is the retention of the devices? endurance?

The "forgetting" experiment is described in a way that is not straightforward enough.

Homeostasis is an important idea and should be introduced in body text, not only in a figure caption and in methods. It would be useful to compare (based on the idea, no quantified results necessary) with the rule used in Ref 18.

D. Appropriate use of statistics and treatment of uncertainties

----- < BR>

Although the field does not require systematic statistics, here, the analysis of the results is sometimes a little imprecise. "we obtained fairly consistent convergence points for the remaining three runs", "This may indicate that even 10k events might have been insufficient to achieve convergence given the choices of biasing parameters" "...show a clear preference" "a substantially clearer classification". All these sentences would be stronger if they were backed by a quantitative analysis (percentage of divergence, difference in classification values...).

E. Conclusions: robustness, validity, reliability

I wish the discussion were longer. Maybe it is an opportunity to include a little neuroscience? (this is a suggestion)

F. Suggested improvements: experiments, data for possible revision

"This is achieved despite the generally noisy and slow drift of the devices' resistive states away from their initial values (panel (c)),"

I really wish more data analysis was done to quantify noise and drift in the learning experiment. This would be important for the field.

G. References: appropriate credit to previous work?

The voltage pulses used for STDP (Fig 1(a)) were originally proposed in Ref 18 and this should be acknowledged.

The Jo et al reference mentioned above is an important milestone for the use of STDP with memristors and should be highlighted in the introduction.

Similarly these visionary Snider works should be acknowledged:

Snider, G. S. (2008, June). Spike-timing-dependent learning in memristive nanodevices. In *Nanoscale Architectures, 2008. NANOARCH 2008. IEEE International Symposium on* (pp. 85-92). IEEE.

Snider, G. S. (2007). Self-organized computation with unreliable, memristive nanodevices. *Nanotechnology*, 18(36), 365202.

H. Clarity and context: lucidity of abstract/summary, appropriateness of abstract, introduction and conclusions

The paper is clear and reads very well.

Reviewer #2 (Remarks to the Author):

##Reviewer Comments

#General Comments

This paper presents a metal-oxide-based synapse for unsupervised, competitive learning using a hybrid hardware/software setup.

The paper addresses an important question in neuromorphic engineering, with relatively clean but somewhat sparse data.

It is remarkable that changes in the devices are gradual in both directions, as most two-terminal resistive devices such as PCM and metal-oxide-based ones often lacked this property in one of the two directions. This result alone is was previously presented in some of the authors earlier work [23]. The results are a stepping stone towards a system that is capable of solving a problem of more practical scale that would be of interest for the more general research community.

The methods section describing the fabrication of the device does not contain any references to the authors previous work, and how the current device is similar to the one proposed in [23], for instance. As a result, it is difficult for the reader to find the key reference for the devices and some key information, such as device area and operation voltage is missing. I cannot find the values of V_{th} and Fig1a does not have a vertical scale.

Discussion of other neural network implementations using the multilevel range of two terminal

resistive devices (other than the authors') are missing: Eg. Eryilmaz et al. 2014 Frontiers in Neuroscience, and Burr et al. 2015 IEEE Transactions on Electron Devices.

The size of current and projected TiO₂ devices compared to SRAM cell in a modern and future CMOS processes might be helpful to underline the importance of this work. A brief sentence or two on peripheral circuits (non-)necessities (selectors, etc.) would be helpful too.

A critical shortcoming of this manuscript is that the presented data seems to be the results of a single run of the respective experiments. The authors do not demonstrate or discuss the robustness and trial-to-trial variability of the learning. This problem is compounded by the fact that only a small portion of the presented network is in hardware. The authors should conduct experiments showing the variability and robustness in their devices.

Due to the hybrid design of the experimental setup, the results do not study the dynamical interplay of plasticity dynamics and network dynamics. Thus it is more a demonstration of the metal-oxide-based synapse rather than a demonstration of a probabilistic neural network with such synapses (as the title implies). See also comment on P7 below. Related to this is that the importance of the role of the homeostatic mechanism on the results is not made clear enough. At one point, the authors state that its role is minor. But the reader then wonders why it was included in the first place, especially knowing that homeostatic plasticity is not straightforward to implement in hardware due to the long time constants involved.

In conclusion, my opinion on this manuscript in its current form and content is that it is more suitable for a specialty journal.

#Specific comments

Fig1b: there is a small but visually perceptible slope on pre-only events. The authors should comment on the sharpness of the threshold and the consequences of this spurious drift. More quantitative results are needed rather than a visual inspection offered by Fig S4.

P5: "Recent rigorous analyses ... input patters." The following references [33,] should probably be moved

P6: Discuss robustness to pattern overlap and the hardware cost in mitigating it.

P6: Parameters are extracted from conductances for mapping onto the software neurons. This raises questions on real interfacing with analog neurons. Namely, is such a mapping necessary and what will the cost of amplifying, shifting and bounding these conductances be?

P7: "Homoeostatic plasticity thus plays only a minor role in the overall learning process". It is not clear how the authors arrive to this conclusion.

P7: "On the other hand the software synapses, being inherently more well-controlled than their hardware counterparts experience a gentle drift towards their final state."

The successful learning of the preference for pattern 0110 in the hardware synapses may be due to

the presence of the well-controlled software synapses. The singling-out of one neuron through WTA dynamics and software synapse preference learning may greatly simplify the task of the hardware synapses in learning the remaining pattern. My initial concern is the stochastic dynamics of the memristors. The authors should comment on this.

One way to overcome this concern is to use software synapses weight that follow a stochastic process that whose variability is more comparable to those of the hardware synapses.

Fig 3ef: The authors should add error bars on the fire count.

P8: The second set of experiments needs to be better explained. Namely, for which reason the software neurons needs to be reinitialized? I assume this is because a trained network is very unlikely to switch its pattern preference. Adding to the confusion is this "as to render their neuron even better suited". "even" is misleading here, since software synapses were not specialized for 0110 beforehand.

P8: "their tendency to experience noisy and abrupt changes in resistive state". The abruptness of the synapse changes has not been discussed.

P18: spurious characters "t*-ime of"

Acronyms in references are smallcase (enclose bibtex entries with curly brackets)

Reviewer #3 (Remarks to the Author):

Probabilistic neural networks with multi-state metal-oxide memristive synapses

by A. Serb, J. Bill, A. Khat, R. Berdan, R. Legenstein, T. Prodromakis

Designing and studying artificial neural networks based on memristive devices in respect learning abilities and brain-like functions represent highly interesting multidisciplinary field with a great potential. The present manuscript reports on unsupervised learning networks using TiO₂-based memristive devices.

The manuscript is well written and the experiments are carefully conducted. It should be considered for publication. However, there are several issues that require to be addressed prior to a decision on the manuscript. Two points appear especially important to me - the proper justification of sufficient elements of novelty (to merit publication in Nature Communication), and the appearing situation that the TiO₂ memristor is regarded as a black box and no relation between the observed neuromorphic functionality and the physical processes within the memristor devices are established or at least discussed. More detailed comments are provided below.

Comments on the text:

1. The novelty elements should be properly explained and justified for the readers. Several recent publications report on similar behaviour of similar systems based on memristive devices are not

discussed or cited. For example Yang et al. Adv Mater 27 (2015) 7720 or Du et al. Adv Funct Mater 25 (2015) 4290. Also the difference with the concepts presented in references 20-26 should be clearly made.

2. In biological systems (live cells) the activity of synapses and neurons based on ionic equilibria are quite well understood. In contrast, in the present manuscript the TiO₂- memristors are considered as a black box. It should be avoided that the message to the readers would be understood as: "We don't know why and how, but it works somehow". A deeper understanding is required on the relation between individual cell properties (in a sense of physics and chemistry) and the observed network functionality. Even if not all details are possible to be clarified at least reliable discussion should be provided. The Chua's memristor is a passive circuit element. How this is related to the active behavior of the Pt/TiO₂/Pt devices?

3. The authors are discussing on membrane potentials. How they define them in Pt/TiO₂/Pt? How these potentials are formed and how they behave? This is comment related to point 2.

4. I have some doubts that the used Pt/TiO₂/Pt devices are reliable for long term operation and good reproducibility. Most of the recent research is orientated to other oxide systems, bilayer cells and asymmetric devices such as Ta/Ta₂O₅/Pt or Hf/HfO₂/Pt. To my knowledge no applications based on TiO₂ memristors are reported. The authors should comment on this. Of course for a proof of concept this point is not essential.

Technical comments:

How the conductivity presented in the figures is measured or calculated?

Which equipment is used to apply and read the pulses?

What is the device size?

Please, provide an equivalent schema of the circuit used for the measurements.

What are the measured resistances in Fig. 3 (used for determining the weights)?

Response to Reviewers

Reviewer #1:

A. Summary of the key results

In this work, Serb and coauthors investigate the use of memristors as artificial synapses that learn using a biology inspired learning rule (STDP). The paper associates theoretical analysis, experiments and simulations in a convincing and rare way. In particular, a small system is investigated experimentally.

We thank the reviewer for their endorsement of our research methodology. We fully agree with the idea that experimental backing of the very interesting neuroscience ideas is of key importance to the field of bio-inspired electronics (especially using memristors).

B. Originality and interest

Learning with memristive synapses has already been done, but with simpler learning rules (eg Ref 24). Using the more sophisticated STDP is an idea that interests the field considerably. STDP has been done experimentally with memristors only in simplistic visions on single devices. Much deeper studies have been done, but in simulation only. This is the first work that validates experimentally, on a very simple system, the use of STDP with memristors. The comparison with simulations is insightful: experiments appear much more noisier than simulations, but the system still learns, due to the robustness of STDP. This is what was believed would happen, but had never be shown experimentally. Of course, in this work, the experimental learning is still very modest. But I believe that the work is of high interest to the field, if some revisions are made.

We thank the reviewer for acknowledging the novelty and niche of our work.

The paper states that Ref 24 uses binary memristors. I believe this to be incorrect, Ref 24 is also using analogue memristors. Ref 24 is an important milestone. Instead of "a form of deterministic, supervised learning", I would prefer that the authors name the learning approach that Ref 24 is using.

Indeed the observation is correct. We apologise for this oversight. The perceived binarity of the synapses probably resulted from Fig. 4b where we can observe that the bulk of the learning seems to occur on the very first epoch. Of course this is no proof of device binarity and we have removed this remark from the introduction.

Furthermore, the learning approach is now explicitly stated as the Manhattan update rule, which is a version of the perceptron rule. The corrections can be found on Ins 46-59.

C. Data & methodology: validity of approach, quality of data, quality of presentation

The paper is well written.

We thank the reviewer for the kind remark.

Figure 3 is too hard to interpret (especially Fig 3(c) that is critical to the message). Maybe the authors could split Fig 3 into several Figures?

Thanks to this round of reviewing and the constructive feedback provided by all referees, we have performed additional experiments that we feel has substantially advanced the quality of our experimental data. As such, all figures are updated accordingly and simplified as much as possible for

aiding the reader. We hope the new way of displaying results has also improved clarity in Fig. 3. Splitting it into more subfigures was considered, but in the end we could not find a satisfying multiple-figure solution.

The discussion that nonlinearity in the device is useful for unsupervised learning is important. "Such dependence of conductance changes on the actual memristive state has commonly been observed in memristors" should be backed by references, eg: Jo, S. H., Chang, T., Ebong, I., Bhadviya, B. B., Mazumder, P., & Lu, W. (2010). Nanoscale memristor device as synapse in neuromorphic systems. Nano letters, 10(4), 1297-1301.

The same dependence on initial state has also been observed in phase change memories, eg in: Suri, M., Bichler, O., Querlioz, D., Traoré, B., Cueto, O., Perniola, L., ... & DeSalvo, B. (2012). Physical aspects of low power synapses based on phase change memory devices. Journal of Applied Physics, 112(5), 054904.

We thank the reviewer for this remark. In particular pointing out that it is not just MOx-based but also PCM-based devices that can exhibit some of the key characteristics that enabled the behaviour observed in this work is important for illustrating the potential of memristors that come in many guises and making this work relevant to a broader audience. The relevant references have been added in the results section Ins 100-102.

The discussion that ends in "In other words, the memristive synapse should be able to encode and store in its resistive state the conditional probability $p(P \text{ RE} | \text{POST} = 1)$ " is interesting and insightful. Actually, a lot of discussion about this already appears in a review paper: Querlioz, D., Bichler, O., Vincent, A. F., & Gamrat, C. (2015). Bioinspired programming of memory devices for implementing an inference engine. Proceedings of the IEEE, 103(8), 1398-1416. Equations (5)-(7) of this review show exactly the points made by the authors, with a similar STDP rule. The authors should rephrase the article to take benefit of the analysis in this review.

We thank the reviewer for making us aware of this particular reference and noting its reference to our work. Following the reviewer's remark, we have now amended the relevant text. We also note that it is quite interesting that both MOx and PCM memories seem to have at least in principle an exponential or exponential-like dependence of weight change vs. initial weight when subjected to repeated fixed-waveform (e.g. fixed amplitude/duration pulse) stimulation. The changes and relevant reference are at Ins 104-107.

What is the retention of the devices? endurance?

We appreciate that retention and endurance are key performance indicators for practical applications and have now included the relevant data in the supporting material as Figs. S5, S6 along with describing the methodology for acquiring this (Ins 743-754).

The "forgetting" experiment is described in a way that is not straightforward enough.

A review of the section indeed reveals that more explicit explanation is in order. In the revised

manuscript a proper introduction to the logic behind the forgetting experiment is given and consolidated as a self-standing paragraph. The precise way in which this was applied in the 2 test runs under discussion is then discussed in a separate paragraph. The corrections can be found on Ins 269-274 when the forgetting experiment is introduced.

Homeostasis is an important idea and should be introduced in body text, not only in a figure caption and in methods. It would be useful to compare (based on the idea, no quantified results necessary) with the rule used in Ref 18.

As suggested by the referee, the role of homeostasis in learning is now briefly explained in the main text Ins 199-203 and the contribution of ref. 18 has been acknowledged. Furthermore, a qualitative comparison with the rule used in ref. 18 has been given in the methods section on Ins 469-474 (WTA network set-up III; homeostatic plasticity section).

D. Appropriate use of statistics and treatment of uncertainties

Although the field does not require systematic statistics, here, the analysis of the results is sometimes a little imprecise. "we obtained fairly consistent convergence points for the remaining three runs", "This may indicate that even 10k events might have been insufficient to achieve convergence given the choices of biasing parameters " "...show a clear preference" "a substantially clearer classification". All these sentences would be stronger if they were backed by a quantitative analysis (percentage of divergence, difference in classification values...).

We appreciate that meaningful quantitative metrics could provide a more satisfying way for conveying the messages in the snippets above; the following points are aimed at addressing this remark:

- 1) **Convergence point consistency:** On Ins 147-151 and Fig. 2 we now pool and fit the converged conductance vs. LTP/LTD composition data to a fitting function and quote the resulting root-mean-square error as a means of quantifying the consistency of convergence across runs.
- 2) **Incomplete convergence hypothesis:** This was an important point that proved interesting to investigate. We have clarified our statement that incomplete convergence might have played a role in causing the differences between the different runs in Fig. 2(a) on Ins 153-158, and added supporting material Fig. S12 and the related paragraph on page 35 where we try to extrapolate the behaviour of each memristor and determine whether the extrapolated conductance convergence point allows a more clear link between LTP/LTD composition and converged conductance to emerge.
- 3) **Preference of hardware-synapse neuron to pattern 0110:** The preference of hardware synapses for pattern 0110 in the initial WTA network run is now evidenced by defining the notion of 'specialisation' Ins 206-210 and examining how the gap between the two patterns widens over the run quite substantially.
- 4) **Classification quality:** Specialisation of the neurons on different input patterns is now directly quantified through the specialisation metric In 239. We quote classification success rate, i.e. state how much % of the time the correct neuron responds to its assigned pattern.

E. Conclusions: robustness, validity, reliability

I wish the discussion were longer. Maybe it is an opportunity to include a little neuroscience? (this is a suggestion)

We have taken the referee's suggestion on board and have expanded the discussion section with some words on Ins 356-393 clarifying the links to experimental and theoretical neuroscience further, as well as tackling the role of homeostatic plasticity within the framework of this manuscript.

F. Suggested improvements: experiments, data for possible revision

"This is achieved despite the generally noisy and slow drift of the devices' resistive states away from their initial values (panel (c))," I really wish more data analysis was done to quantify noise and drift in the learning experiment. This would be important for the field.

We thank the referee for this comment and have now added a note on how we quantified this in the main text on Ins 247-254, and proceed to add text on page 33 in the supporting material as well as table T6 and figure S11, where we explained how this quantification was carried out. Fundamentally we fitted the evolution of weights over trials to fitting functions, computed the noise as the standard deviation of the residuals and then made a simple attempt to correlate this to observed measurement noise levels as well as the observable noise levels that arise as a result of the randomness of the input signal. The results, we believe were quite interesting as summarized in table T6.

G. References: appropriate credit to previous work?

The voltage pulses used for STDP (Fig 1(a)) were originally proposed in Ref 18 and this should be acknowledged. The Jo et al reference mentioned above is an important milestone for the use of STDP with memristors and should be highlighted in the introduction. Similarly these visionary Snider works should be acknowledged:

Snider, G. S. (2008, June). Spike-timing-dependent learning in memristive nanodevices. In Nanoscale Architectures, 2008. NANOARCH 2008. IEEE International Symposium on (pp. 85-92). IEEE.

Snider, G. S. (2007). Self-organized computation with unreliable, memristive nanodevices. Nanotechnology, 18(36), 365202.

We thank the reviewer for helping making the link to previous work more complete. The points above are now reflected in the updated manuscript. All references have been introduced/had their contributions acknowledged. (Snider references now in the introduction, Ins 32-35).

H. Clarity and context: lucidity of abstract/summary, appropriateness of abstract, introduction and conclusions

The paper is clear and reads very well.

We thank the reviewer for the remark. We take the time investment of reviewers and readers alike

very seriously and we hope that this current improved version will further aid the reader to appreciate the contribution of this work, unhindered by any poor explanation.

Reviewer #2:

##Reviewer Comments

#General Comments

This paper presents a metal-oxide-based synapse for unsupervised, competitive learning using a hybrid hardware/software setup.

The paper addresses an important question in neuromorphic engineering, with relatively clean but somewhat sparse data. It is remarkable that changes in the devices are gradual in both directions, as most two-terminal resistive devices such as PCM and metal-oxide-based ones often lacked this property in one of the two directions. This result alone was previously presented in some of the authors earlier work [23]. The results are a stepping stone towards a system that is capable of solving a problem of more practical scale that would be of interest for the more general research community.

We thank the referee for acknowledging the novelty and importance of our work.

The methods section describing the fabrication of the device does not contain any references to the authors previous work, and how the current device is similar to the one proposed in [23], for instance. As a result, it is difficult for the reader to find the key reference for the devices and some key information, such as device area and operation voltage is missing. I cannot find the values of V_{th} and Fig1a does not have a vertical scale.

With regard to the information on fabrication, we typically provide the fabrication details of specific devices for each manuscript, as these experience change over time, reflecting on the continuing improvement of our processes. However, in the majority of our work all devices come from the same basic 'family'. In response to this and other reviewers' requests for more information on the devices, the methods section has been complemented with additional information on the devices fabrication.

The devices employed in this work are a slight variation on those utilised in our previous work [25, 51], now stated on Ins 409-410. With regard to device area we now mention that our devices were fabricated with optical lithography and are thus in the micrometer scale and as such far from their full downscaling potential. This is explicitly stated on Ins 401-402. Using these devices allowed us to better concentrate on demonstrating reliably how complex synaptic-like behaviour based on memristive devices can be harnessed in practice. The threshold voltages for all devices used in this work are now summarised in the supporting material, table T1 with a description of how these are defined on page 23. Fig. 1a is intended as a schematic diagram of the pulsing scheme that can be used to generate LTP and LTD with both pre- and post-signal time-of-arrival dependence. The x-axis has the specific timing settings used in this work, which are common to all devices whilst the y-axis is left qualitative because the specific voltage levels are tailored to each device. We now point out that

the threshold values and LTP/LTD induction parameters are shown the supporting material. This is done in the caption to Fig. 1.

Discussion of other neural network implementations using the multilevel range of two terminal resistive devices (other than the authors') are missing: Eg. Eryilmaz et al. 2014 Frontiers in Neuroscience, and Burr et al. 2015 IEEE Transactions on Electron Devices.

We thank the reviewer for pointing out these crucially important publications. These have now been included along with some key information about their achievements on Ins 46-59.

The size of current and projected TiO2 devices compared to SRAM cell in a modern and future CMOS processes might be helpful to underline the importance of this work. A brief sentence or two on peripheral circuits (non-)necessities (selectors, etc.) would be helpful too.

The prospect of practical implementation and what such endeavour will entail is indeed important. We now discuss the points above extensively on Ins 324-355.

A critical shortcoming of this manuscript is that the presented data seems to be the results of a single run of the respective experiments. The authors do not demonstrate or discuss the robustness and trial-to-trial variability of the learning. This problem is compounded by the fact that only a small portion of the presented network is in hardware. The authors should conduct experiments showing the variability and robustness in their devices.

The reviewer has picked up a very valid point that often plagues memristor research. To this we took the following action:

- We repeated all learning experiments.
- The 'free, unsupervised learning' from Fig. 3 has been repeated 3 times with appropriate device initialisations each time and showed an ability to both start from a naïve state and to segregate the prototype patterns by the end of the epoch. Results are summarised in Fig. S8 whilst the experiment is explained in its own section of the supporting material titled 'repeatability of learning' on page 29.
- The learning reversal experiments have also been repeated, with an extra run at the end (in the supporting material, Fig. S9 and experiment description in a dedicated supporting material section titled 'learning reversibility timescale check' on page 32) where we demonstrate much clearer reversal capabilities than in the previous data-set.
- Added endurance and retention data on our devices showing reliability at least in the thousands of switching cycles and stability over at least 2.5 hours (overnight runs show this extending to at least 15 hrs – not shown here). These are summarised in Figs. S5, S6 and related text on page 26.

Due to the hybrid design of the experimental setup, the results do not study the dynamical interplay of plasticity dynamics and network dynamics. Thus it is more a demonstration of the metal-oxide-based synapse rather than a demonstration of a probabilistic neural network with such synapses (as the title implies). See also comment on P7 below. Related to this is that the importance of the role of the homeostatic mechanism on the results is not made clear enough. At one point, the authors state that its role is minor. But the reader then wonders why it was included in the first place, especially knowing that homeostatic plasticity is not straightforward to

implement in hardware due to the long time constants involved.

In conclusion, my opinion on this manuscript in its current form and content is that it is more suitable for a specialty journal.

In order to address this interesting point we have introduced imperfection factors in the software synapses that bring them much closer to memristor behaviour. This is mentioned in the main text on lns 214-216, covered in the methods section on page 15 and beyond under its own heading titled 'WTA network set-up II; adding realistic imperfections in software synapse function' and demonstrated through added results and analysis in the supporting material; specifically Fig. S11, table T6 and the discussion on page 33. This way we have managed to keep the relevance to the general community by demonstrating that a system operating both synapses that are and that behave similarly to memristors does lead to encouragingly reliable learning with the capability of complete learning reversal if needed. The specific interplay between plasticity and network dynamics is a key aspect of this work and has been covered through simulation work, but up to date no experimental evidence has shown any device family (and associated electronic system) capable of such complex behaviour. Given the enormous rift between theoretical/simulation work and practical achievements (very little work goes beyond single synapse demonstration as a rule with some notable exceptions –Strukov and IBM groups for example-), we believe that this is a truly major milestone, if for no other reason than that it demonstrates physically that a neuromorphic system concept (the network presented in this work) placing extremely stringent requirements on the memristors operating it can be attacked through memristor technology in hardware. The shift of such complicated system from the theoretical to the practical is something that our understanding of the community suggests a large number of people doubt is even possible.

With regard to homeostasis, it is true that we mentioned the effects are not significant, however, that was a vestige from a previous version of the document that we forgot to delete after changing the relevant section. We apologise for this oversight as this statement is not true and we appreciate that it stands completely out of context. We have now removed the offending text and added a few words on the true influence of homeostatic plasticity in the discussion section, on lns 381-393.

Finally, we have updated the title of the manuscript to “**Unsupervised learning** in probabilistic neural networks with multi-state metal-oxide memristive synapses” to better reflect our original contributions.

#Specific comments

Fig1b: there is a small but visually perceptible slope on pre-only events. The authors should comment on the sharpness of the threshold and the consequences of this spurious drift. More quantitative results are needed rather than a visual inspection offered by Fig S4.

The sharpness of the memristors' threshold voltages are indeed an often overlooked aspect of practical device behaviour which we had left out for brevity. However, following the comment above we realised that it is of interest for further practical work. As such we now added into the discussion section, lns 311-323 a part where we note how the softness of the threshold is related to the contrast between the invasiveness of the pre-only, pre+post and post-only, which in turn will affect

the degree of unwanted plasticity induced by the pre-only waveform as well as the degree to which the peak and trough of the post waveform exert concurrent control over DUT RS. In addition this has also been linked to the effects of spurious drift on learning, (again Ins 311-323). The drift observed in the 'neutral regions' of figures S1 and S4 has also been quantified in the supporting material on pages 23 and 26, Ins 738-742 and table T2. The subject of drift and its complicated parameter space (stimulus shape, running DUT RS-dependence and the 3-way relationship between pre-, post-peak and post-trough voltage levels) are in our opinion interesting and challenging enough to warrant a dedicated study and consolidation in a corresponding (specialist) publication.

P5: "Recent rigorous analyses ... input patters." The following references [33,] should probably be moved

We spent some time wondering whether these references should be in the introduction or not and in the end it seemed more readable if the references mentioned above were bundled together in a 'mini-introduction' for the neural network part of the experiment alone. This allows the focus to stay on the memristor synapses (whose demonstration was the main aim of this work) during the entire introduction and then shift focus to how the neural network was set-up to leverage them to the wide extent that we have shown in this work.

P6: Discuss robustness to pattern overlap and the hardware cost in mitigating it.

Note that the input pattern sets as defined show overlap in patterns. The prototype patterns 1001 and 0110 were presented as noisy versions where each component was independently inverted with a probability of 10%. Hence, the presented patterns for prototype 1001 included patterns 0001, 1000, 1101, 1011. These patterns were denoted by 1001_ Δ (analogous noisy versions of 0110 were denoted by 0110_ Δ) and responses of the system to them were analysed in the manuscript, see e.g. Fig. 3(e),(f). Our results show that the system is very robust in this respect since those neurons that specialised on the prototype also responded to the corresponding Δ patterns after learning. Note that in particular the noisy versions 1101 and 1011 show significant overlap with the other prototype 0110 since they include one of their two non-zero bits. It is worth mentioning that the WTA architecture is in general very robust to such overlaps. For the current setup, we did not use pattern overlap in the prototype patterns because of their very low dimensionality. The theory for WTA networks and experience from computer simulations (see e.g. ref. [44] in the revised manuscript) show that such overlap poses no difficulties for the circuit for high-dimensional inputs. Hence we do not expect any additional hardware cost to account for pattern overlap due to the inherent robustness of WTA circuits to such pattern sets. We added a brief discussion of this interesting point on Ins 367-380.

P6: Parameters are extracted from conductances for mapping onto the software neurons. This raises questions on real interfacing with analog neurons. Namely, is such a mapping necessary and what will the cost of amplifying, shifting and bounding these conductances be?

The key point to make here is that if a future neuromorphic/memristive system operates using the biasing schemes described in this work, then each neuron will have to be equipped with a waveform-shaping output circuit. We point out that this should not be a particular problem if the

waveforms involved are square waves of variable durations because of the ease of implementation using digital clock signals (we have an idea on how to do this, but that would dig into too much detail for a non-specialist journal like Nat. Comms.) and the cost of these waveform-shapers is at 1/neuron rather than 1/synapse. Of course there may be (for high voltage memristors only) the added inconvenience of laying out 1-3 more power rails and including those pins in the pad-ring with extra ESD protection etc. etc. but that really does require a more dedicated study. The changes are on Ins 341-352.

On a related note a current issue with the memristive synapses used in this work is that they did operate optimally only after the voltages applied to them for LTP and LTD were individually tailored. Despite the fact that the differences in voltage levels between devices are fairly small (see table T4) the ambition is that improved control during fabrication and electroforming will eventually deliver devices that operate with sufficiently uniform LTP and LTD voltages. Already in T4 we see that 2 of the devices used had the exact same voltage requirements. A brief remark to that end is included on Ins 352-355 as part of the discussion.

P7: "Homeostatic plasticity thus plays only a minor role in the overall learning process". It is not clear how the authors arrive to this conclusion.

We thank the reviewer for pointing us to this sentence. This was an unfortunate statement that was forgotten during final editing – and makes little sense without its previous context in the figure. We have removed the statement from the caption of Figure 3, and we have added a few sentences in the Discussion section on the general contribution of the homeostatic intrinsic plasticity rule to unsupervised learning in the WTA network (Ins 381-393).

P7: "On the other hand the software synapses, being inherently more well-controlled than their hardware counterparts experience a gentle drift towards their final state." The successful learning of the preference for pattern 0110 in the hardware synapses may be due to the presence of the well-controlled software synapses. The singling-out of one neuron through WTA dynamics and software synapse preference learning may greatly simplify the task of the hardware synapses in learning the remaining pattern. My initial concern is the stochastic dynamics of the memristors. The authors should comment on this. One way to overcome this concern is to use software synapses weight that follow a stochastic process that whose variability is more comparable to those of the hardware synapses.

We sincerely thank the referee for this excellent point that allowed us to improve our case. We have now repeated all experiments with software synapses this time corrupted by two different kinds of imperfections: variability in the amount each synapse weight changes given identical initial conditions (reflecting memristor cycle-to-cycle variation) and a term corrupting the weight values used for computation by the network reflecting measurement noise observed by our instrument. This is mentioned in the main text on Ins 214-216, covered in the methods section on page 15 and beyond under its own heading titled 'WTA network set-up II; adding realistic imperfections in software synapse function' and demonstrated through added results and analysis in the supporting material; specifically Fig. S11, table T6 and the discussion on page 33.

Fig 3ef: The authors should add error bars on the fire count.

This is indeed a good point in that repeat experiments should give an idea of how variable the system is. We feel however that showing a set of 3 experiments of free, unsupervised learning and include each set of results in full could be much easier for an interdisciplinary audience to follow. This was done to give more detail, but also because the number of times each input pattern is presented to the network within a particular interval varies itself. As such, error bars would be 'contaminated' by this extra variation and might end up being slightly misleading. Full results in Fig. S8

P8: The second set of experiments needs to be better explained. Namely, for which reason the software neurons needs to be reinitialized? I assume this is because a trained network is very unlikely to switch its pattern preference. Adding to the confusion is this "as to render their neuron even better suited". "even" is misleading here, since software synapses were not specialized for 0110 beforehand.

The reason for the software synapse initialisations in the subsequent neural network runs is now explained more explicitly on Ins 269-278, 282-283, 285-286, 290-291. The phrasing now should also eliminate the confusion caused by the term 'even' previously. Having some software synapses enabled us to set-up the experiment in such way and control the flow of plasticity in the memristors a bit more directly; justifying our experimental methodology.

P8: "their tendency to experience noisy and abrupt changes in resistive state". The abruptness of the synapse changes has not been discussed.

The comment on the abruptness of switching was referring more to the fact that the noise is 'jagged', i.e. prominent at quite high frequencies as opposed to e.g. an obvious 50Hz component. We realise, however that this can be confusing so we deleted the remark.

P18: spurious characters "t*-ime of". Acronyms in references are smallcase (enclose bibtex entries with curly brackets)

These have now been rectified.

Reviewer #3:

Probabilistic neural networks with multi-state metal-oxide memristive synapses

by A. Serb, J. Bill, A. Khiat, R. Berdan, R. Legenstein, T. Prodromakis

Designing and studying artificial neural networks based on memristive devices in respect learning abilities and brain-like functions represent highly interesting multidisciplinary field with a great potential. The present manuscript reports on unsupervised learning networks using TiO₂-based memristive devices.

The manuscript is well written and the experiments are carefully conducted. It should be considered for publication. However, there are several issues that require to be addressed prior to a decision on the manuscript. Two points appear especially important to me - the proper justification of sufficient elements of novelty (to merit publication in Nature Communication), and the appearing situation that the TiO₂ memristor is regarded as a black box and no relation between the observed neuromorphic functionality and the physical processes within the memristor devices are established or at least discussed. More detailed comments are provided below.

We thank the referee for the endorsement of our work and we hope our response below fully addresses the provided comments.

Comments on the text:

1. The novelty elements should be properly explained and justified for the readers. Several recent publications report on similar behaviour of similar systems based on memristive devices are not discussed or cited. For example Yang et al. Adv Mater 27 (2015) 7720 or Du et al. Adv Funct Mater 25 (2015) 4290. Also the difference with the concepts presented in references 20-26 should be clearly made.

The novelty elements in this work are explained much more explicitly now on Ins 62-70. The relation with existing literature is also more clearly expressed.

2. In biological systems (live cells) the activity of synapses and neurons based on ionic equilibria are quite well understood. In contrast, in the present manuscript the TiO₂- memristors are considered as a black box. It should be avoided that the message to the readers would be understood as: "We don't know why and how, but it works somehow". A deeper understanding is required on the relation between individual cell properties (in a sense of physics and chemistry) and the observed network functionality. Even if not all details are possible to be clarified at least reliable discussion should be provided. The Chua's memristor is a passive circuit element. How this is related to the active behavior of the Pt/TiO₂/Pt devices?

The question of how these memristors actually work inside is one of the key open problems in the field. This work does not seek to address this issue, coming directly from an application perspective based on an understanding of practical behaviour and supported by the fact that our memristors actually exhibit enough structure in their responses to input stimulation to enable such experiments

to succeed. Nevertheless, since this is such an important aspect of the problem, a tentative link to materials science and electrochemistry is attempted in the supporting material, page 36 and beyond as a self-standing section titled 'materials level interpretation' on a working-hypothesis basis. We explain key device behaviours as such:

-Gradual switching: filament formation/destruction by gradual migration of ions/vacancies. Lower voltages cause slower drift of fewer ions/vacancies thus resulting in more gradual switching and longer endurance.

-Increased variation in converged conductances when scrambling LTP/LTD mixture order: Abruptness of change in pulsing strategy affects overall aggressiveness of ion/vacancy motion in the material. Possible explanation.

-Drift: nanobattery effect affecting primarily HRS region. Effect not particularly destructive for current application.

3. The authors are discussing on membrane potentials. How they define them in Pt/TiO₂/Pt? How these potentials are formed and how they behave? This is comment related to point 2.

Within the context of this work the notion of a 'membrane potential' is defined exclusively at the neuron level. Each neuron responds to input patterns by generating an output that depends on the input pattern and the synaptic weights through which it is processed. We name this output the 'membrane potential' in analogy to biological neurons. That 'membrane potential' then drives the spiking behaviour of the neuron. However, once the neuron has spiked, the presence of the spike is encoded into an appropriate bias voltage waveform for the memristor-synapse to receive either as a pre- or as a post-type event, which will then drive the synapse. The value we defined as membrane potential is completely unrelated to the voltage used to drive the device. Throughout the manuscript we use the term 'membrane potential' with its first meaning, i.e. from the point of view of the neuron, rather than the synapse. This is now explicitly stated on Ins 195-199.

4. I have some doubts that the used Pt/TiO₂/Pt devices are reliable for long term operation and good reproducibility. Most of the recent research is orientated to other oxide systems, bilayer cells and asymmetric devices such as Ta/Ta₂O₅/Pt or Hf/HfO₂/Pt. To my knowledge no applications based on TiO₂ memristors are reported. The authors should comment on this. Of course for a proof of concept this point is not essential.

We concur with the referee that when it comes to stability, other metal oxide systems are considered more useful for memory applications. Simultaneously, however, the results coming out of our TiO₂ devices and our experience in working with them rendered them a good material for performing this proof-of-concept-level work. We now mention the beneficial characteristics of TiO_x-based devices that enticed us to use them as test subjects on Ins 301-310 and include retention and endurance data in the supporting material, figures S5, S6 to indicate that despite not being able to match HfO_x and TaO_x devices in memory-related performance, TiO_x is still a sufficiently good material for interesting applications.

Technical comments:

How the conductivity presented in the figures is measured or calculated?

Which equipment is used to apply and read the pulses?

Please, provide an equivalent schema of the circuit used for the measurements.

Conductance values indicated throughout the paper are measured by our own in-house-developed instrument. This is now stated in the on Ins 493-495 and in the supporting material on page 36. A simplified scheme of the circuitry used to measure conductances is now also included as Fig. S13.

What is the device size?

The devices used in this work were manufactured using optical lithography and are therefore in the micrometer scale. This approach has allowed us to concentrate on the proof-of-concept aspects of this work relating to memristor operation within the network. Nevertheless the prospect of downscaling is important and the logical next step is to attempt the experiment with nano-scale devices and understand how the additional issues introduced by scaling such as increased line resistance may affect performance. We now clearly state that the experiments were performed on micrometer-scale devices on Ins 401-402.

What are the measured resistances in Fig. 3 (used for determining the weights)?

The mapping between conductance values and weights is summarized in table T4 in the supporting material. We now make that clear to readers directly where the conductance-weight mapping is addressed for the first time in the main text on Ins 226-228.

Reviewers' comments:**Reviewer #1 (Remarks to the Author):**

The authors have largely improved the manuscript, especially the supplementary material. They have well addressed my questions and comments, by adding interesting new results and discussion.

The software synapses are now noisy, as the hardware ones, which makes sense. However, I really miss the old learning Figure where we could see the difference between perfect software synapses that learnt in a very clean way, and the hardware synapses that learnt in a noisy way, but still succeeded at learning. It was very demonstrative, and I highly suggest reincluding this, at least in supplementary information.

The new statement about homeostasis:

"its implementation in neuromorphic designs comes surprisingly cheap since it only requires a local accumulator circuit per neuron"

I don't believe this to be true (as pointed by one of the reviewers). Homeostasis requires long time constants: i.e. big counters or big caps; it is possible to do in hardware, but does not come cheap.

"The reasons behind this are unclear, especially given the difficulty in extrapolating memristor behaviour into the future. The possibility that even 10k events might have been insufficient to

achieve convergence given the choices of biasing parameters (± 2.5 V) can therefore not be excluded, even though some basic checks do not immediately lend support to this hypothesis (see supporting material, Fig. S12)"

This sounds like a loose end that could be solved by a few numerical simulations and theoretical analysis. Maybe it just does not converge well (which would not undermine the paper's conclusions). This is actually quite interesting point, as real life data will be very noisy.

Although I love the paleontology analogy, calling the early days of memristor research a "Cambrian explosion" might be an exaggeration.

Reviewer #2 (Remarks to the Author):

The authors have done an excellent job in addressing my comments. The results presented in this manuscript are much more convincing.

A few residual and minor points remain to be addressed:

Micrometer-scale devices:

"our devices were fabricated with optical lithography and are thus in the micrometer scale and as such far from their full downscaling potential"

Downscaling will be critical for practical implementations. Discuss whether/how downscaling might affect the properties of the device.

I would have wished that the issue that the devices "operate optimally only after the voltages applied to them for LTP and LTD were individually tailored." would be explained more explicitly in the discussion around line 352. Namely, if the devices need to be tailored individually, this would entail 1/synapse devices. Relating to the point above, wouldn't downscaling the devices deteriorate matters more than what improvements over fabrication and electroforming can provide? If yes, then this should be discussed.

Spell out DUT RS in the main text, or just use DUT resistive state as in the the rest.

Reviewer #3 (Remarks to the Author):

In their responses that authors have addressed my comments and correspondingly amended the manuscript. In my view a discussion on the physical processes related to the functionality in the main text would be of great advantage. However, I acknowledge that the focus of the manuscript is made on the functionality of the network and not on the physical principles of operation. I can therefore recommend the manuscript for publication.

Response to Reviewers:

Reviewer #1:

The authors have largely improved the manuscript, especially the supplementary material. They have well addressed my questions and comments, by adding interesting new results and discussion.

We thank the reviewer for their endorsement of our modifications.

The software synapses are now noisy, as the hardware ones, which makes sense. However, I really miss the old learning Figure where we could see the difference between perfect software synapses that learnt in a very clean way, and the hardware synapses that learnt in a noisy way, but still succeeded at learning. It was very demonstrative, and I highly suggest reincluding this, at least in supplementary information.

We have now added the requested figure (S8) as well as the related synaptic evolution figure (S9) in the supporting material for completeness. Together they show what happens if noisiness in the software synapses is disabled. The stark contrast at both neuron and individual synapse level behaviour is underlined.

The new statement about homeostasis:

"its implementation in neuromorphic designs comes surprisingly cheap since it only requires a local accumulator circuit per neuron"

I don't believe this to be true (as pointed by one of the reviewers). Homeostasis requires long time constants: i.e. big counters or big caps; it is possible to do in hardware, but does not come cheap.

We appreciate the reviewer's concern and have accordingly removed any mention of a 'cheap' hardware implementation. We now state that it is in principle 'possible' instead as indicated by the referee. Line 406.

"The reasons behind this are unclear, especially given the difficulty in extrapolating memristor behaviour into the future. The possibility that even 10k events might have been insufficient to achieve convergence given the choices of biasing parameters (± 2.5 V) can therefore not be excluded, even though some basic checks do not immediately lend support to this hypothesis (see supporting material, Fig. S12)"

This sounds like a loose end that could be solved by a few numerical simulations and theoretical analysis. Maybe it just does not converge well (which would not undermine the paper's conclusions). This is actually quite interesting point, as real life data will be very noisy.

We appreciate this as being an interesting point, which could not, however, be fully addressed throughout this work. The reason behind this is that it requires substantial developments in memristor modelling with sufficient predictive power, an area that we are actively engaged in. We have now rephrased the corresponding passage to reflect this more clearly, Ins. 159-162:

“Attempting to extrapolate memristor behaviour by exponential fitting, as presented in supporting material, Fig. S4 indicates that even 10k events seem insufficient to achieve convergence given the choice of biasing parameters (also see Supplementary note 5). We believe that this could be potentially addressed as more realistic memristor models appear.”

Although I love the paleontology analogy, calling the early days of memristor research a "Cambrian explosion" might be an exaggeration.

Addressed accordingly. Phrase removed altogether.

Reviewer #2:

The authors have done an excellent job in addressing my comments. The results presented in this manuscript are much more convincing.

We thank the reviewer for their endorsement.

A few residual and minor points remain to be addressed:

Micrometer-scale devices:

"our devices were fabricated with optical lithography and are thus in the micrometer scale and as such far from their full downscaling potential"

Downscaling will be critical for practical implementations. Discuss whether/how downscaling might affect the properties of the device.

Memristors are indeed not immune to downscaling. We have now explicitly stated this in the discussion section, Ins. 341-343:

“Even though the performance of memristor devices is also known to be impacted by downscaling, through for example increased access wire resistance, the advantage over SRAM is expected to dominate.”

I would have wished that the issue that the devices "operate optimally only after the voltages applied to them for LTP and LTD were individually tailored." would be explained more explicitly in the discussion around line 352. Namely, if the devices need to be tailored individually, this would entail 1/synapse devices. Relating to the point above, wouldn't downscaling the devices deteriorate matters more than what improvements over fabrication and electroforming can provide? I yes, then this should be discussed.

The issue is now explicitly stated in the discussion section, Ins. 365-371:

“Notably, in this work biasing conditions were individually tailored for each memristive synapse, a result of device-to-device variability which is expected to become increasingly challenging with downscaling. Improvements in control over fabrication and electroforming conditions are needed to counterbalance that effect and deliver memristors that operate under sufficiently uniform biasing

conditions in order to use a single, non-programmable waveform shaping circuit for all devices in practical systems.”

Spell out DUT RS in the main text, or just use DUT resistive state as in the the rest.

Replaced by ‘resistive state’ across entire text.

Reviewer #3:

In the their responses that authors have addressed my comments and correspondingly amended the manuscript. In my view a discussion on the physical processes related to the functionality in the main text would be of great advantage. However, I acknowledge that the focus of the manuscript is made on the functionality of the network and not on the physical principles of operation. I can therefore recommend the manuscript for publication.

We thank the reviewer for their endorsement of the changes made.